# Development of Durum Wheat Breads Low in Sodium Using a Natural Low-Sodium Sea Salt

**DOI:** 10.3390/foods9060752

**Published:** 2020-06-05

**Authors:** Elena Arena, Serena Muccilli, Agata Mazzaglia, Virgilio Giannone, Selina Brighina, Paolo Rapisarda, Biagio Fallico, Maria Allegra, Alfio Spina

**Affiliations:** 1Di3A—Dipartimento di Agricoltura, Alimentazione e Ambiente, University of Catania, via S. Sofia 100, 95123 Catania, Italy; earena@unict.it (E.A.); agata.mazzaglia@unict.it (A.M.); selina.brighina@unict.it (S.B.); bfallico@unict.it (B.F.); 2CREA—Consiglio per la ricerca in agricoltura e l’analisi dell’economia agraria, Centro di Ricerca Olivicoltura, Frutticoltura e Agrumicoltura, Corso Savoia 190, 95024 Acireale (Catania), Italy; serenamuccilli@hotmail.com (S.M.); paolo.rapisarda@crea.gov.it (P.R.); maria.allegra@crea.gov.it (M.A.); 3DSAAF—Dipartimento di Scienze Agrarie, Alimentari e Forestali, University of Palermo, Viale delle Scienze, Ed. 4, 90128 Palermo, Italy; virgiliogiannone@hotmail.com; 4CREA—Consiglio per la ricerca in agricoltura e l’analisi dell’economia agraria, Centro di Ricerca Cerealicoltura e Colture Industriali, Corso Savoia 190, 95024 Acireale (Catania), Italy

**Keywords:** *Triticum turgidum* L. subsp. *durum* Desf., bread, NaCl, low-sodium sea salt, Na^+^ reduction, physico-chemical and textural attributes, sensory evaluation

## Abstract

Durum wheat is widespread in the Mediterranean area, mainly in southern Italy, where traditional durum wheat breadmaking is consolidated. Bread is often prepared by adding a lot of salt to the dough. However, evidence suggests that excessive salt in a diet is a disease risk factor. The aim of this work is to study the effect of a natural low-sodium sea salt (Saltwell^®^) on bread-quality parameters and shelf-life. Bread samples were prepared using different levels of traditional sea salt and Saltwell^®^. The loaves were packaged in modified atmosphere conditions (MAPs) and monitored over 90 days of storage. No significant differences (*p* ≤ 0.05) were found in specific volumes and bread yield between the breads and over storage times, regardless of the type and quantity of salt used. Textural data, however, showed some significant differences (*p* ≤ 0.01) between the breads and storage times. 5-hydroxymethylfurfural (HMF) is considered, nowadays, as an emerging ubiquitous processing contaminant; bread with the lowest level of Saltwell^®^ had the lowest HMF content, and during storage, a decrease content was highlighted. Sensory data showed that the loaves had a similar rating (*p* ≤ 0.05) and differed only in salt content before storage. This study has found that durum wheat bread can make a nutritional claim of being “low in sodium” and “very low in sodium”.

## 1. Introduction

There is much evidence suggesting that excessive salt intake endangers our health [1,2,3], and reducing its consumption is one of the first steps to preventing noncommunicable diseases [4]. Dietary habits are often developed during childhood [5,6,7], so nutritional education towards a low-sodium diet with adequate potassium intake should be encouraged [8,9]. In Italy, salt consumption by children and adolescents suggests that the average daily sodium consumption exceeds the official recommendations [10].

Natural foods contain modest amounts of sodium [11], and approximately two-thirds of salt intake come from its addition during food preparation [12]. Eighty food categories were identified as significant contributors to salt intake, and targets were set for the food industry to meet in each category within a certain period [13].

The WHO member states have agreed to reduce the global population’s intake of salt by a relative 30 % by 2025, and several strategies have been undertaken to improve the consumer’s understanding of healthy eating recommendations [14,15,16,17,18,19].

Nutrition claims of “low sodium/salt”, “very low sodium/salt”, and “sodium/salt-free” for foods containing 1.2, 0.4, and 0.05 g kg^−1^ of sodium, respectively, (or the equivalent value for salt) on food labels, informs consumers about salt content [20,21,22].

Salt is an essential ingredient in breadmaking: it retards gas production, enhances bread flavor, affects the rheological properties of dough, controls fermentation (decreasing yeast activity in the dough), and it can affect the quality parameters of bread [23,24]. Furthermore, NaCl has a strengthening effect on gluten, increasing its resistance or elasticity, and decreasing the extensibility of the dough [25,26].

The strategies to reduce sodium in bread include the use of reduced-sodium sea salt [27], the partial replacement of sodium chloride with potassium chloride and yeast extract [28], the use of a salt substitute with 57% of sodium chloride [29], and heterogeneous NaCl distribution, leading to enhanced saltiness by taste contrast [30].

In bread wheat (*Triticum aestivum* L.), salt is generally used at levels of about 1–2% based on flour weight [31]. A survey of salt content in artisan and industrial bread produced in all Italian regions was conducted in 2009/2010, its data having been recently published [32]. Artisan breads contained between 0.7% and 2.3% g/100 g of salt, while industrial bread, on average, contained 1.6% salt, most samples (56%) having a very high content. In the Mediterranean area, the cultivation of durum wheat (*Triticum turgidum* L. subsp. *durum* Desf.) is widespread compared to that of bread wheat [33] as it has a greater tolerance to drought, high temperatures, and fungal diseases, but less resistance to winter and spring cold. According to traditional uses, mainly in Southern Italy, bread is prepared from remilled durum wheat semolina [34]. Durum wheat milling products are characterized by peculiar chemical, rheological, colorimetric, and baking properties [35,36,37,38,39]. The bread has a compact texture, being in some cases excessively dense, with lower specific volume and harder crumbs than white bread [39], and the characteristic taste and flavor are generally enhanced by adding a high percentage of salt, from 20 to 25 g kg^−1^ [28].

In the last few years, the use of low-sodium salts in foods has been recommended.

However, almost all the commercially available low-sodium salts are produced by blending purified potassium chloride with ordinary table salts to achieve a reduced sodium content.

Natural low-sodium sea salt not only provides less sodium and does not affect the taste profile, but contains lots of essential trace minerals such as magnesium, potassium, calcium, and other nutrients the body requires.

Therefore, the aim of this paper is to evaluate the effect of substituting salt with low-sodium sea salt to measure the quality parameters of durum wheat bread over long storage.

## 2. Materials and Methods

### 2.1. Materials

Durum wheat (*Triticum turgidum* L. subsp. *durum* Desf.) remilled semolina for breadmaking [39] was provided by “Valle del Dittaino Società Cooperativa Agricola” (Assoro, Italy), an industrial bakery with a durum wheat mill.

The bread ingredients were food grade. Compressed yeast (AB Mauri, Casteggio, Italy) and traditional sea salt (99.5% NaCl; Mulino S. Giuseppe, Catenanuova, Italy) employed in the breadmaking process were purchased in a local retailer. Saltwell^®^ (Salinity Group, Saltwell AB Göteborg Sweden) is a natural low-sodium sea salt (less than 35%) extracted from an underground sea below the Atacama desert (Chile). This natural sea salt contains 65 ± 1% NaCl, 30 ± 1.5% KCl, 1.0 ± 0.1% of MgSO_4_, 0.5 ± 0.1% of CaSO_4_, and traces of other salts and minerals. Saltwell^®^ was kindly donated by Medsalt—Mediterranean Salt Company S.r.l. (Rome, Italy).

Various levels (1.70%, 0.35%, 0.15% on semolina basis) of traditional sea salt and Saltwell^®^ were used in dough formulations, as listed in Table 1.

### 2.2. Methods

#### 2.2.1. Physico-Chemical and Rheological Analyses of Remilled Semolina

The physico-chemical analyses of remilled semolina were carried out following the methods indicated by Giannone et al. (2018) and Palumbo et al. (2002) [39,40,41]: moisture content was determined according to the AACC 08-01 method (AACC, 2000) [42]. Protein content was determined by means of the Infratec 1241 Grain Analyzer (Foss Tecator, Höganäs, Sweden), based on near infrared transmittance. Ash content was determined according to the AACC 44-19 method (AACC, 2000) [42]. The particle size distribution was determined by a LabSifter (KBF7SN, Buhler, Switzerland). Remilled semolina for breadmaking was sieved for exactly 5 min on sieves with openings of 300, 200, 180, and 160 μm. Wet and dry gluten and gluten index were obtained by using a Glutomatic System (Glutomatic 2200, Centrifuge 2015, Glutork 2020; Perten Instruments AB, Huddinge, Sweden), according to the UNI 10690 method (UNI, 1979) [43]. The α-amylase activity was obtained by using the Falling Number 1500 apparatus (Perten Instruments AB, Huddinge, Sweden), according to the ISO 3093:2009 method (ISO, 2009) [44].

The CIELAB color parameters (*L**, *a**, *b**) were determined by Chromameter CR-300 (Minolta, Osaka, Japan), using the illuminant D_65_. Alveograph indices were determined according to the AACC method 54-30A (AACC, 2000) [42] using an alveograph model MA 87 equipped with the software Alveolink NG (Tripette et Renaud, Villeneuve-la-Garenne, France). Farinograph parameters were obtained according to the AACC 54-21 method (AACC, 2000) [42] using a Farinograph, equipped with the software Farinograph^®^ (Brabender instrument, Duisburg, Germany).

#### 2.2.2. Bread Sample Production and Packaging

Bread samples were produced in a local breadmaking company (“Valle del Dittaino Società Cooperativa Agricola”, Assoro, Italy), according to a proven industrial formulation: remilled durum wheat semolina (65 kg), compressed yeast (0.9% on semolina basis), water (66.0% on semolina basis) and the corresponding amount of salt. Six bread formulations containing different levels of traditional sea salt and low-sodium sea salt were produced (Table 1). The dough was mixed for 17 min in a high-speed mixer (San Cassiano, Italy). The final dough temperature was 26 ± 1 °C. The dough was left to rest in bulk for 15 min, divided into 980 ± 20 g portions (100 loaves for each production), proofed for 150 min at 32 ± 1 °C and 66 ± 2% relative humidity (RH) and baked at 220 °C for 60 min, in industrial tunnel ovens measuring 33 × 3 m (Pavailler Engineering, Galliate, Italy). The baked loaves, with an approximate weight of 1 kg each, were automatically transported to a cooling chamber (Tecnopool, Italy) set at 20 ± 2 °C for 120 min. After cooling, the loaves were sliced by means of an automatic slicing machine (Brevetti Gasparin, Marano Vicentino, Italy) to 11 ± 1 mm thickness. About 450 g of sliced bread per loaf was packaged under modified atmosphere conditions (MAPs) using inert gas (70:30 N_2_:CO_2_). The bread packaging materials consisted of two plastic films provided by Cryovac Sealed Air (Elmwood Park, NJ, USA).

The samples were stored for up to 90 days at 20 ± 2 °C and 60 ± 2% RH. The quality parameters were determined at regular intervals in triplicate for each batch.

The following parameters and properties were tested for each bread sample during each sampling: volume, height, weight, diameter basis, crumb porosity, internal structure, top and base crust thickness, texture profile analysis, water activity, moisture, pH, 5-hydroxymethylfurfural (HMF) content, crust and crumb color, and sensory evaluation.

#### 2.2.3. Bread Quality Evaluation

##### Determination of the Physico-Chemical Properties of the Breads

The volume was determined in a loaf volume meter by measuring the volume of rapeseed displaced by the bread, according to the AACC method 10.05.01 (AACC, 2000) [42]. The specific volume (mL/g) was calculated as a ratio of the loaf volume and the bread weight. The specific weight was calculated as the ratio of the loaf weight and bread volume. The h/d ratio was obtained as the ratio of the bread height and bread diameter of the loaf base. The crumb porosity was estimated using the Mohs scale. The CIELAB space *L* a* b** color parameters were measured for the crumb, in the transversely cut bread, and on the crust surface, averaging ten distinct points in each case, using a chromameter (CR-200, Konica Minolta, Osaka, Japan) with illuminant D_65_.

Bread samples were analyzed for Na^+^ (mg Kg^−1^) content by inductively coupled plasma optical emission spectrometry (ICP-OES Optima 2000DV, Perkin Elmer, Italy). The samples were first ground to a powder, and oven-dried at 105 °C for 4 h until constant weight, then an aliquot equal to 0.5 g was weighed and placed in a muffle furnace at 600 °C for 12 h. After mineralization, the ashes were dissolved in 4 mL of distilled water and 0.5 mL of nitric acid at 69.5% (Superpure; Merck, Darmastadt, Germany). The solutions were poured into 50 mL flasks and brought to volume with distilled water before the analyses.

Water activity (a_w_) was determined by a Hygropalm 40 AW (Rotronic Instruments Ltd., Crawley, UK) according to the manufacturer’s instructions. Three bread slices (11 ± 1 mm thickness) were used, after removal of the crust. For each set of determinations, separate loaves were used.

The moisture content of bread crumb was determined by oven drying at 105 °C until constant weight, according to AOAC method no. 945.15 [45]. The pH was measured according to [46] using a pH meter (Mettler Toledo, MP 220).

### 2.3. Texture Profile Analysis of Breads

The texture profile analysis (TPA) of bread was determined using a Universal testing machine (model 3344, Instron, Norwood, MA, USA.) equipped with a cylindrical probe of 50 mm of diameter and a 2000 N load cell. Data were acquired through Bluehill^®^ 2 software (Instron, Norwood, MA, USA). Cyclic compression tests (a 30-s gap between first and second compression) were set up: the crosshead speed was 3.3 mm/s, the force required to compress the samples by 40% was recorded on 5-cm side square portions of 24-mm thick slices, and the average value of five replicates was taken. The TPA profile recorded four primary parameters: hardness (N), springiness (mm), resilience, gumminess, and one derived parameters (chewiness, N mm).

### 2.4. HMF Extraction and HPLC Analysis

HMF was extracted and determined following the methodology proposed by [28]. Ground bread samples (5 g; La Moulinette, Moulinex, 2002) and 25 mL of water (J.T. Baker, Deventer, Holland) were put into a volumetric flask (50 mL) and stirred for 10 min. Then the sample was diluted up to 50 mL with water (JT. Baker, Deventer, Holland) and centrifuged for 45 min at 5000 rpm. An aliquot of the supernatant was filtered through a 0.45-μm filter (Albet) and injected into an HPLC system (Shimadzu Class VP LC-10ADvp) equipped with a DAD (Shimadzu SPD-M10Avp). The column was a Gemini NX C18 (150 × 4.6 mm, 5 μm; Phenomenex) fitted with a guard cartridge packed with the same stationary phase. The HPLC conditions were the following: isocratic mobile phase, 90% water (J.T. Baker) at 1% acetic acid (Merck), and 10% methanol (Merck); flow rate, 0.7 mL/min; injection volume, 20 μL. The wavelength range was 220–660 nm, and the chromatograms were monitored at 283 nm. HMF was identified by splitting the peak of the HMF from the bread-solution sample with a standard of HMF (*p* > 98% Sigma-Aldrich, St. Louis, MO, USA) and by comparing the UV spectra of the HMF standard with that of the bread samples. All analyses were performed in duplicate, including the extraction procedure, and the reported HMF concentration was, therefore, the average of four values. The results were expressed as mg of HMF per kilogram of dry matter.

### 2.5. Sensory Evaluation

The sensory profile [28,47] was defined by a trained [48] panel of 12 judges (six females and six males, 28–40 years old). The judges, recruited for their individual abilities, had more than five years of experience in the sensory analysis of bread and bakery products, and they were submitted to further training over 4 weeks to generate attributes using handmade and industrial breads and to familiarize themselves with the scales and procedures. The judges, using a discontinuous scale between 1 (absence of the sensation) and 9 (extremely intense), have evaluated the intensity of the 11 sensory attributes selected on the basis of frequency (≥60%), following the definitions given by [49,50,51] (Table 2).

The evaluation sessions, performed at 0, 15, 30, 60, and 90 days of storage, were conducted in the sensory laboratory [52] of Di3A (University of Catania, Italy) from 11:00 a.m. to 12:00 a.m. in individual booths illuminated with white light. The sliced bread samples were served on plates, coded with three-digit numbers, and water was provided to judges for rinsing between samples. The order presentation was randomized among judges and sessions using a randomized complete block. All data were acquired by a direct computerized registration system (FIZZ Biosystems. ver. 2.00 M, Couternon, France).

### 2.6. Statistical Analysis

The statistical analysis was performed using the Statgraphics^®^
*Centurion XVI* software package (Statpoint Technologies, INC.). A two-way analysis of variance (ANOVA), followed by Tukey’s HSD test (*p* ≤ 0.001; *p* ≤ 0.01; *p* ≤ 0.05), was carried out on physico-chemical and textural attributes. The data were expressed as means ± standard deviations. The sensory data for each attribute were submitted to one-way ANOVA. The significance was tested by means of the F-test. A principal component analysis (PCA) was performed using PAST, Paleontological Statistics software package, 2011 [53].

## 3. Results and Discussion

### 3.1. Physico-Chemical and Rheological Characterization of the Durum Wheat Remilled Semolina

Physico-chemical characteristics of remilled semolina were moisture 13.8 ± 0.07 g/100 g, protein 12.2 ± 0.10 g/100 g, and ash 0.87 ± 0.01 g/100 g. These quality parameters met the Italian legal requirements [54]. Particle size distribution was >300 µm: 11.0 ± 1.73 g/100 g; between 200–300 µm: 26.3 ± 1.15 g/100 g; between 180–200 µm: 22.0 ± 2.00 g/100 g; between 160–180 µm: 20.0 ± 1.00 g/100 g; <160 µm: 20.7 ± 4.04 g/100 g. These findings agreed with those reported by other authors for remilled semolina [39]. Dry gluten content was 10.0 ± 0.1 g/100 g. The gluten index value was 80.7 ± 4.0, and the value of amylase activity at the falling number was low (577 ± 3.0 s). Regarding dry gluten content and relative qualitative index, the sample exhibited regular gluten quantities and high gluten tenacity. Similar values were reported by [28,39,55].

As regards color parameters, the values were lightness (*L**) 71.0 ± 0.3, red index (*a**) - 2.12 ± 0.02, yellow index (*b**) 18.52 ± 0.05.

Rheological behavior was evaluated by alveograph. Deformation energy (W) was 209 ± 4 10^−4^ × J, while the tenacity/extensibility (P/L) value showed a tenacious dough (value = 2.5). Strong gluten is expected in remilled durum wheat semolina [24].

Mixing behavior was evaluated by a farinograph apparatus. The semolina sample indicated the quantity of water absorbed at 500 BU (Brabender Unit), and the dough consistency was 60.6 ± 0.04% due to high protein content. The values of dough development time (1 min, 48 s ± 3.0 s), dough stability (4 min ± 12 s), and softening index (58 ± 1 BU) agreed with those reported by other authors on remilled semolina [28,38,39,55].

### 3.2. Sodium Content in Bread

The levels of the two salts used in the loaves, the sodium content, and the minimum limits established by EU regulations [20,21] applying to nutritional claims are shown in Table 3.

### 3.3. The Quality Parameters of Breads and Their Evolution during Storage

The *p*-values for all the physical and textural parameters of the bread types with respect to storage time are reported in Table 4.

The specific volumes and weights of the loaves were significant for each of the two factors of variability (type (A), storage time (B), and their interaction (A × B), even with different *p* levels (*p* ≤ 0.001 for storage time, *p* ≤ 0.01 A × B interaction, and *p* ≤ 0.05 per type; see Table 4).

The results of the physical and textural properties of the industrial breads in the MAP conditions during 90 days of storage are shown in Table 5 and Table 6.

No significant differences in specific volumes were shown among the bread samples, regardless of the type and level of sea salt (Table 4).

These findings agree with those reported by [23], but they disagree with those reported by [24]. Additionally, no significant differences in specific weight were observed among the controls and other bread samples or during storage time. The addition of different types and quantities of sea salt did not decrease bread yield. After 60 days of storage, the specific weight decreased.

The ratio between the height and diameter of the loaves used in the baking industry to parametrize possible dough failure was significant (*p* ≤ 0.001) for all the factors and their interaction (Table 4). At time 0, control A was found to have the greatest h/d ratio (approximately 4.5) due to the addition of ordinary sea salt (Table 5). The other bread samples, as expected, showed a lower ratio during storage, especially the bread samples containing less traditional sea salt and sea salt with reduced Na^+^. These findings agree with those reported by [28].

Significant differences were found for loaf porosity among the types (*p* ≤ 0.001) and the A × B interaction (*p* ≤ 0.05), but not for storage time (B) (Table 4). After baking (t0), almost all the types, except for 2B, showed proper development of crumb porosity. Starting from 15 days of storage, the performance of 2A also slightly decreased (Table 5).

Significant differences were found between the types (*p* ≤ 0.001) and storage times (*p* ≤ 0.01 and *p* ≤ 0.001, respectively) but not for A × B interaction as regards internal structure and top crust thickness (Table 4). As for internal structure, only control A had an irregular structure over the whole storage time. Similar results were reported by [28].

As for top crust thickness, for up to 30 days of storage, no remarkable differences were recorded among the types (mean value of 3.8 mm); after 60 days, the values decreased up to 2.67 mm for control B.

No significant difference was highlighted for basis crust thickness between the types, the different storage times, and their interactions (Table 4). Almost all the bread samples exhibited a mean value of basis crust thickness of 4 mm. These findings agree with those reported by [28].

Three of the five parameters of texture profile analysis (hardness, gumminess, and chewiness) were always significant (*p* ≤ 0.001), while resilience and springiness were significant per type and storage time (*p* ≤ 0.001), but not for A × B interaction (Table 4). The two control breads (1A and 1B), as expected, showed lower values for the first three parameters. Starch retrogradation (i.e., the recrystallization of polysaccharide in gelatinized starch) is believed to be the main cause of crumb firmness change during storage [56].

Textural data highlighted high values of hardness, with significant differences among the samples, as reported by [39], and storage time (Table 6).

The hardness values, as expected, increased as the storage period progressed. As regards the bread samples, control A reported the lowest values during the entire storage period. Up to t30, the two controls, albeit with statistically different values, recorded the lowest hardness values. From t60, the control A values remained low, while the control B values increased until reaching about 55 N at the end of storage.

No significant differences in springiness or resilience were shown among the bread samples and during the storage times, whatever the type and level of salt (Table 6). Up to 30 days of storage, no remarkable differences were recorded among the breads (mean value of 5.7 mm); after 60 days, the values of springiness increased by up to 7.0 mm. These findings do not agree with those reported by [39].

As for resilience, the average value was around 0.80. During the entire storage period, the two controls showed higher resilience values. From the end of the baking to the end of storage, resilience values decreased slightly. These findings agree with those reported by [39].

With regard to gumminess and chewiness, they increased progressively with increasing storage times and with decreasing salt content, regardless of type, until they reach the maximum at t90 for 2B (58.0 and 426.0). During the entire storage time, the two controls always showed the lowest values, and were similar to each other, except for t90.

Water activity (a_w_) and moisture content were significant compared to all the factors of variability (Table 7). As for pH and HMF, they were significant compared to all the factors of variability (*p* ≤ 0.001; Table 7).

Crumb lightness and redness were significant compared to all the factors of variability. Crumb yellowness was significant for bread (A) and storage time (B) (*p* ≤ 0.001) but not for their interaction (A × B) (Table 7). The effect of the addition of sea salt with reduced Na^+^ on the *L** parameter of crumb during the entire time storage was not significant (Table 7).

Chemical properties of the breads during the storage time are reported in Table 8.

Crumb a_w_ is an important parameter of food processing and conservation technologies that comes into play for food stability and safety. It indicates the amount of free water not linked by bonds with the soluble constituents of the food, i.e., the water that can participate in chemical, physical, biological, and enzymatic reactions.

In general, water activity is a relatively easy parameter to measure, which can be an advantage, especially in the food industry [57].

The a_w_ value ranged from about 0.88 for Control A at t90, to 0.93 for 2A at t0 (Table 7). Similar values have been reported by [55].

After baking, and up to t15, there is no difference among the breads. From t30, water activity decreases for both controls. From t60 to the end of storage, a_w_ decreases slightly for all the types. At t90, only the a_w_ value of Control A is lower than the other types. Moisture content ranged from about 35.5–38.4% at the beginning (Table 8). Bread samples containing natural low Na^+^ sea salt show the highest moisture content, and significant differences were found between all the breads. During storage, the breads with NaCl generally show the highest levels of moisture, and at 90 days of storage, the moisture content decreased, ranging from 35.3–32.4%. No significant differences were found between control B and samples 1B (1.22% and 0.25% Saltwell^®^) and the bread samples with the lowest levels of salt (2A and 2B).

The pH ranges from 5.36 to 5.93 at the beginning; at 90 days of storage, it ranges from 5.73 to 5.82 (Table 8). The variability seems to be more related to the storage time rather than to the different levels and salts used in the recipe. Similar trends were reported both for durum wheat bread with yeast extract and fortified with fiber [28,50].

HMF is a widely used compound as heat induces the chemical index generally used for monitoring thermal abuse [58,59,60,61]. In bread and in other baking products, HMF is used to monitor the heating process, and several factors influence its formation, such as manufacturing conditions and recipe [57,58,59]. Even if the toxicity risk of HMF is still debated, nowadays, HMF is under evaluation as an emerging ubiquitous processing contaminant since there is evidence to suggest that HMF and its metabolites may have harmful effects on human health [60,61,62,63].

Among foods, coffee and bread contribute the most HMF exposure, about 85% of total intake [64].

The HMF parameter was significant compared to all the factors of variability (*p* ≤ 0.001; Table 7). HMF levels at the beginning ranged from about 23 to 39 mg/kg of dry matter (Table 8), and significant differences were found between all samples. These levels were lower than those reported for durum wheat bread with KCl and taste enhancer [28], and it is known that differences in water content in the leavening and/or baking time and the ratio between crumb and crust of the loaf could influence HMF content [58]. Bread samples with the lowest levels of natural low Na^+^ sea salt (2 B) had the lowest HMF content. During storage, a decrease in HMF amount was highlighted, though the trend in decrease was not regular. Generally, the bread samples with the lowest levels of salt had the lowest HMF content due to the effects of a high level of NaCl on starch degradation and yeast growth, resulting, in both cases, in higher levels of Maillard indicators [65]. At 90 days of storage, this parameter ranged from about 20.6 to 25.5 mg/kg of dry matter. The HMF trend during storage was similar to those reported by [28,50], suggesting that HMF decrease is more related to storage time rather than recipe.

During storage, crumb redness in the traditional sea salt (control A) test slowly decreased. After t15, the *a** value begins to decrease for all breads (Appendix A).

### 3.4. Sensory Evaluation

The addition of different types and quantities of sea salt had little effect on the sensory characteristics of the bread sample. Table 9 reports the ANOVA results of sensory data and the bread attributes, which significantly differentiated at different *p*-levels (*p* ≤ 0.05; *p* ≤ 0.01; *p* ≤ 0.001), at each sampling. Mean values were reported only for significantly different attributes.

At t0, the bread samples were evaluated similarly by panellists, with the exception of the “salty” attribute. Obviously, the control breads (Control A and Control B) had the highest value of saltiness.

At 15 and 30 days of storage, the samples were significantly different for the attributes sweet, salty, bread flavor, and overall evaluation. The 0.15 NaCl sample showed the highest intensity of sweet taste, while the control samples, as expected, had the highest score of salt, bread flavor, and overall evaluation.

At 60 and 90 days of storage, the attributes of sweet, salty, and overall significantly differentiated the bread samples. The 0.15 NaCl and 0.15 Saltwell^®^ bread samples had the highest intensity of sweet and the lowest of the attributes salt and overall. The control samples showed the highest intensity of the attribute overall.

The different levels of sea salt did not influence the attributes of texture (i.e., softness), as reported by [28].

Table 10 reports the sensory attributes which significantly differentiated (*p* ≤ 0.05) during the 90 days of storage.

Control A showed a significant decrease during storage but only for the attributes of humidity and softness. At 0 and 15 days of storage, Control A had the highest intensity of these two sensory attributes.

Control B showed a significant decrease during storage for the attributes of elasticity, humidity, and softness. These attributes began to decrease after 30 days of storage.

Sample 2A showed a significant decrease only for the attribute humidity, while bread samples 1A, 1B, and 2B did not show any significant differences during the 90 days of storage.

During storage, the bread samples did not develop off-odors or off-flavors in agreement with those reported by [28].

### 3.5. Multivariate Statistical Analysis

Principal component analysis (PCA) is a multivariate analysis that allows the reduction and interpretation of large multivariate datasets with some underlying linear structure. In this trial, it was carried out to determine if and which salt (type and concentration) had an influence on the qualitative and sensory traits of the breads. The PCA included the following 24 dependent variables: specific volume, specific weight, h/d ratio, crumb porosity, hardness, gumminess, chewiness, springiness, resilience, water activity, moisture, pH, HMF, acidity, and crust and crumb color parameters (as *L**, *a**, *b**, *h*, *C*).

The two main factors accounting for 56.92% of the total variance were PC1 and PC2 at 37.08% and 19.84% (Figure 1).

There are two types of trends on the first axis: (1) based on salt content, the groups shift from the negative to the positive section, from the breads with minimum salt concentrations (2A and 2B), to those with more (Control A and Control B) (Figure 1); (2) based on days of storage, from the longest (t90) to the shortest (t0) (Figure 1). Convex hulls were used to highlight these trends. They can be defined as the intersection of all convex sets containing a given subset of a Euclidean space. The convex hull of a set of data is the smallest convex set that contains it.

The variables that determined these trends were resilience, crust color (as *a**, *C*), h/d ratio, crumb color (*h*), and moisture, which showed the highest positive loading values (0.272, 0.230, 0.227, 0.228, 0.218, and 0.209 respectively), chewiness, hardness, gumminess, and springiness, with the highest negative loadings (−0.314, −0.311, −0.309, −0.263, respectively).

The groups also showed a gradient with respect to the days of storage, if PC2 is observed: from the positive scores of the longer storage time to the gradually lower scores of the shorter ones (Figure 1).

The variables that positively correlated with PC2 were crust color parameters (*L**, *h*, *b**) and moisture (loading values, 0.367, 0.334, 0.271, 0.292, respectively). Moreover, PC2 negatively correlated with a_w_, specific volume, and crust hardness (−0.294, −0.266, −0.248, respectively).

In summary, sorting the data according to the first two axes distributes the groups in relation to the lowest salt concentration with the maximum storage time, and so on, up to the breads with the highest salt concentrations with the shortest storage times.

PCA loadings did not have the necessary strength to affect the net separation of groups, but this seems to support the hypothesis that the different breads and salt concentrations do not lead to substantial differences in the overall qualitative characteristics and acceptability of the product.

## 4. Conclusions

The results of this study showed that replacing traditional sea salt with Saltwell^®^ in durum wheat bread is a possible strategy for reducing sodium intake while maintaining the quality and sensorial characteristics of the bread.

There were no significant differences in the specific volume and bread yield among bread samples and during storage times, regardless of the type and level of sea salt used. The textural data showed high hardness and chewiness values, with significant differences between samples and storage times.

Sensory data showed that the different levels of sea salt did not influence the attributes of softness.

Principal component analysis (PCA) seems to support these findings since, overall, the parameters analyzed were unable to differentiate groups effectively.

Natural low sodium sea salt has made it possible to obtain durum wheat bread with the nutritional claim “low in sodium” (<0.12 g/100 g) and/or “very low in sodium” (<0.04 g/100 g) on food labels, in accordance with EU regulations [20,21,22]. However, the breads showed good taste and flavor.

These results should encourage the opportunity to produce low-sodium or very low-sodium bread because of consumers’ increasing interest in durum wheat bread in accordance with the guidelines for a healthy diet.

## Figures and Tables

**Figure 1 foods-09-00752-f001:**
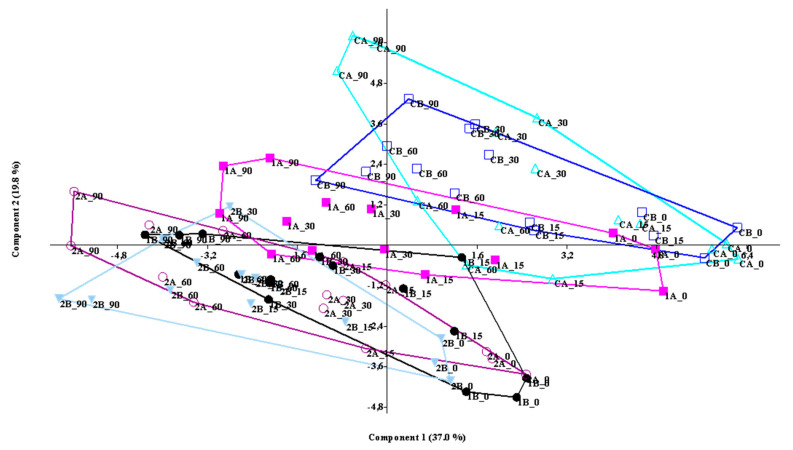
Principal component analysis (PCA) scatter diagram defined by the first two principal components (i.e., PC1, PC2) and convex hulls for the measured physico-chemical and textural traits of the breads, grouped by storage time.

**Table 1 foods-09-00752-t001:** Bread type code and percentage of two salts on remilled semolina basis.

Bread Types Code	Salt Added (% *w*/*w* Remilled Semolina)
Control A	1.70% Traditional sea salt
Control B	1.70% Saltwell^®^
1A	0.35% Traditional sea salt
1B	0.35% Saltwell^®^
2A	0.15% Traditional sea salt
2B	0.15% Saltwell^®^

**Table 2 foods-09-00752-t002:** Descriptive terms used for sensory profiling of bread.

	Attributes	Definition	Scale Anchors	
Crumb appearance	Crumb color	Color intensity of crumb	Whitish	Light yellow
	Alveolar structure	Porosity of crumb	Fine and uniform	Coarse and poorly homogeneous
Visual-tactile	Elasticity	Ability of the crumb to recover from compression exerted by fingers	Slow and partial recovery	Fast and complete recovery
	Humidity	Humidity perceived at the surface of bread crumb	Dry	Humid
Aroma/Flavor	Bread	The typical aroma/flavor of bread just taken out of the oven	Weak	Strong
	Yeasty	The aroma/flavor of a fermented yeast-like	None	Strong
	Wheat	The typical aroma/flavor of wheat	None	Strong
	Off-odour/Off-flavour	Aroma/Flavor unpleasant, not characteristic of bread perceived through taste and smell when swallowing	None	Strong
Taste	Sweet	A basic taste factor produced by sugars	None	Strong
	Salty	A basic taste factor produced by sodium chloride	None	Strong
	Sour	A basic taste factor produced by acids	None	Strong
	Bitter	A basic taste factor produced by caffeine	None	Strong
Mouthfeel	Astringent	Sensory perception in the oral cavity that may include drying sensation and roughing of the oral tissue	None	Strong
Texture	Softness	Force required to compress the product with the molars	Hard	Soft
	Overall evaluation	An overall assessment expressed by considering all of the attributes	Low	High

**Table 3 foods-09-00752-t003:** Percentage of two salts in bread, sodium content and limits established by EU regulations [21,22] (data are means ± standard deviations).

Type	Salt in Experimental Bread (%)	Na^+^ Content (g/100g)	Regulations (EU) No. 1924/2006 and No. 1047/2012—Nutritional Claims
Control A	1.22% Traditional sea salt	0.430 ± 0.014A	-
Control B	1.22% Saltwell^®^	0.240 ± 0.014B	-
1A	0.25% Traditional sea salt	0.087 ± 0.001C	0.12 g of Na^+^—low in sodium
1B	0.25% Saltwell^®^	0.064 ± 0.001C	0.12 g of Na^+^—low in sodium
2A	0.11% Traditional sea salt	0.048 ± 0.001C	-
2B	0.11% Saltwell^®^	0.035 ± 0.000C	0.04 g of Na^+^—very low in sodium

Different capital letters in the same column indicate significant difference (*p* ≤ 0.001).

**Table 4 foods-09-00752-t004:** Analysis of variance of the physical and textural parameters studied on the loaves (*p*-values).

Factors of Variability	Degrees of Freedom	Specific Volume	Specific Weight	h/d Ratio	Porosity	Internal Structure	Top Crust Thickness	Basis Crust Thickness	Hardness	Springiness	Resilience	Gumminess	Chewiness
Type (A)	5	0.002	0.014	0.000	0.000	0.000	0.000	0.444	0.000	0.000	0.000	0.000	0.000
Storage time (B)	4	0.000	0.000	0.000	0.156	0.007	0.000	0.571	0.000	0.000	0.000	0.000	0.000
A × B	20	0.011	0.008	0.000	0.021	0.824	0.064	0.568	0.000	0.529	0.088	0.000	0.000

**Table 5 foods-09-00752-t005:** Evaluation of physical properties during storage of the bread samples produced using different types and levels of sea salt (data are means ± standard deviations).

Days of Storage	Type	Specific Volume (mL/g)	Specific Weight (g/mL)	h/d Ratio	Porosity (1-8) ^a^	Internal Structure (1-2) ^b^	Top Crust Thickness (mm)	Basis Crust Thickness (mm)
0	Control A	3.03 ± 0.09 gh	0.33 ± 0.01 ab	4.46 ± 0.25 a	6.00 ± 0.00 abc	2.00 ± 0.00	3.50 ± 0.00	4.50 ± 0.00
	Control B	3.14 ± 0.20 cdefgh	0.32 ± 0.02 abcd	3.75 ± 0.11 bcd	6.00 ± 0.00 abc	1.00 ± 0.00	3.00 ± 0.00	4.17 ± 0.29
	1A	3.06 ± 0.04 fgh	0.33 ± 0.00 abc	3.60 ± 0.16 bcdefg	6.00 ± 0.00 abc	1.00 ± 0.00	3.17 ± 0.29	4.00 ± 0.00
	1 B	3.09 ± 0.04 defgh	0.32 ± 0.00 abcd	3.58 ± 0.06 bcdefg	6.00 ± 0.00 abc	1.00 ± 0.00	3.83 ± 0.29	4.50 ± 0.50
	2 A	3.09 ± 0.04 efgh	0.32 ± 0.00 abcd	3.23 ± 0.11 defgh	6.00 ± 0.00 abc	1.33 ± 0.58	3.83 ± 0.29	4.67 ± 0.29
	2 B	2.96 ± 0.07 h	0.34 ± 0.01 a	3.11 ± 0.04 gh	7.00 ± 0.00 a	1.00 ± 0.00	4.00 ± 0.00	4.83 ± 0.29
15	Control A	3.20 ± 0.09 cdefgh	0.31 ± 0.01 abcdef	3.77 ± 0.09 bc	5.67 ± 0.00 bc	2.00 ± 0.00	3.50 ± 0.50	4.33 ± 0.29
	Control B	3.13 ± 0.18 cdefgh	0.32 ± 0.02 abcd	3.70 ± 0.12 bcde	6.00 ± 0.00 abc	1.33 ± 0.58	3.17 ± 0.29	5.00 ± 0.00
	1A	3.33 ± 0.03 abcdefgh	0.30 ± 0.00 abcdefg	3.42 ± 0.16 bcdefgh	6.00 ± 0.00 abc	1.00 ± 0.00	3.83 ± 0.29	4.83 ± 0.29
	1 B	3.23 ± 0.23 bcdefgh	0.31 ± 0.02 abcdefg	3.65 ± 0.20 bcdef	6.00 ± 0.00 abc	1.00 ± 0.00	4.33 ± 0.29	4.67 ± 0.29
	2 A	3.35 ± 0.11 abcdefgh	0.30 ± 0.01 abcdefg	3.29 ± 0.27 bcdefgh	6.50 ± 0.00 ab	1.00 ± 0.00	4.33 ± 0.29	5.50 ± 0.50
	2 B	3.43 ± 0.05 abcdefgh	0.29 ± 0.00 bcdefg	3.23 ± 0.04 defgh	7.00 ± 0.00 a	1.00 ± 0.00	4.67 ± 0.29	5.17 ± 0.29
30	Control A	3.33 ± 0.31 abcdefgh	0.30 ± 0.03 abcdefg	3.77 ± 0.22 bc	5.67 ± 0.00 bc	2.00 ± 0.00	3.67 ± 0.00	4.50 ± 0.00
	Control B	3.59 ± 0.04 abc	0.28 ± 0.00 defg	3.83 ± 0.03 b	5.67 ± 0.00 bc	1.67 ± 0.58	3.67 ± 0.29	4.50 ± 0.00
	1A	3.52 ± 0.03 abcdef	0.28 ± 0.00 cdefg	3.39 ± 0.06 bcdefgh	6.00 ± 0.00 abc	1.67 ± 0.58	3.67 ± 0.29	5.00 ± 0.00
	1 B	3.30 ± 0.09 abcdefgh	0.30 ± 0.01 abcdefg	3.27 ± 0.07 cdefgh	6.00 ± 0.00 abc	1.67 ± 0.58	4.33 ± 0.29	4.67 ± 0.29
	2 A	3.45 ± 0.01 abcdefg	0.29 ± 0.00 bcdefg	3.32 ± 0.13 bcdefgh	7.00 ± 0.00 a	1.33 ± 0.58	4.17 ± 0.29	5.50 ± 0.50
	2 B	3.23 ± 0.06 bcdefgh	0.31 ± 0.01 abcdefg	3.34 ± 0.09 bcdefgh	7.00 ± 0.00 a	1.00 ± 0.00	3.83 ± 0.29	4.83 ± 0.29
60	Control A	3.17 ± 0.17 cdefgh	0.32 ± 0.02 abcde	3.62 ± 0.08 bcdefg	5.33 ± 0.00 c	2.00 ± 0.00	3.33 ± 0.58	4.83 ± 0.76
	Control B	3.51 ± 0.19 abcdef	0.29 ± 0.02 bcdefg	3.71 ± 0.09 bcde	5.67 ± 0.00 bc	1.33 ± 0.58	3.33 ± 0.58	5.17 ± 0.29
	1A	3.40 ± 0.10 abcdefgh	0.29 ± 0.01 abcdefg	3.52 ± 0.14 bcdefgh	5.67 ± 0.00 bc	2.00 ± 0.00	3.50 ± 0.50	5.17 ± 0.29
	1 B	3.42 ± 0.05 abcdefgh	0.29 ± 0.00 abcdefg	3.55 ± 0.09 bcdefgh	5.33 ± 0.00 c	1.00 ± 0.00	3.33 ± 0.29	5.17 ± 0.29
	2 A	3.54 ± 0.09 abcde	0.28 ± 0.01 cdefg	3.16 ± 0.16 fgh	7.00 ± 0.00 a	1.67 ± 0.58	3.50 ± 0.50	5.00 ± 0.00
	2 B	3.42 ± 0.08 abcdefgh	0.29 ± 0.01 abcdefg	3.18 ± 0.04 efgh	7.00 ± 0.00 a	1.33 ± 0.58	3.50 ± 0.00	4.50 ± 0.50
90	Control A	3.57 ± 0.19 abcd	0.27 ± 0.03 g	3.49 ± 0.16 bcdefgh	5.33 ± 0.00 c	2.00 ± 0.00	3.00 ± 0.50	4.50 ± 0.00
	Control B	3.54 ± 0.12 abcde	0.28 ± 0.01 cdefg	3.74 ± 0.30 bcd	5.67 ± 0.00 bc	1.33 ± 0.58	2.67 ± 0.58	4.00 ± 0.00
	1A	3.71 ± 0.15 a	0.27 ± 0.01 fg	3.33 ± 0.07 bcdefgh	5.67 ± 0.00 bc	2.00 ± 0.00	3.33 ± 0.58	4.33 ± 0.58
	1 B	3.54 ± 0.15 abcde	0.28 ± 0.01 cdefg	3.40 ± 0.07 bcdefgh	5.33 ± 0.00 c	1.33 ± 0.58	3.17 ± 0.29	4.00 ± 0.00
	2 A	3.68 ± 0.17 ab	0.27 ± 0.01 efg	3.25 ± 0.09 cdefgh	7.00 ± 0.00 a	1.33 ± 0.58	3.17 ± 0.29	4.00 ± 0.00
	2 B	3.28 ± 0.11 abcdefgh	0.30 ± 0.01 abcdefg	3.03 ± 0.25 h	7.00 ± 0.00 a	1.00 ± 0.00	3.17 ± 0.29	3.83 ± 0.29

^a^ 1, most porous; 8, least porous.^b^ 1, regular; 2, irregular. Different letters in the same column indicate significant difference (*p* ≤ 0.01).

**Table 6 foods-09-00752-t006:** Evaluation of textural parameters of bread samples produced using different types and levels of sea salt during storage (data are means ± standard deviations).

Days of Storage	Type	Hardness (N)	Springiness mm)	Resilience	Gumminess	Chewiness (N × mm)
0	Control A	10.57 ± 0.43 o	5.10 ± 0.60	0.91 ± 0.02	9.58 ± 0.31 q	50.36 ± 2.94 o
	Control B	16.97 ± 0.68 mn	4.76 ± 0.46	0.91 ± 0.01	10.26 ± 0.58 pq	52.44 ± 4.99 o
	1A	28.11 ± 0.63 jkl	5.32 ± 0.10	0.85 ± 0.01	20.14 ± 0.32 mno	108.70 ± 0.90 m
	1 B	34.10 ± 2.40 hi	5.12 ± 0.04	0.81 ± 0.01	27.58 ± 2.03 ijk	145.49 ± 1.70 l
	2 A	23.70 ± 3.06 l	6.12 ± 0.22	0.80 ± 0.05	25.72 ± 1.60 ijkl	163.56 ± 2.74 k
	2 B	37.70 ± 0.46 h	5.92 ± 0.58	0.79 ± 0.04	35.42 ± 0.93 h	202.35 ± 4.35 h
15	Control A	25.85 ± 0.14 kl	5.07 ± 0.68	0.87 ± 0.03	14.87 ± 0.60 opq	79.40 ± 1.59 n
	Control B	28.84 ± 0.28 ijkl	5.58 ± 0.39	0.89 ± 0.01	15.53 ± 0.41 nopq	76.20 ± 2.44 n
	1A	47.76 ± 2.36 g	6.26 ± 1.14	0.82 ± 0.03	27.92 ± 1.19 ij	168.22 ± 3.37 ijk
	1 B	34.19 ± 0.78 hi	5.85 ± 0.56	0.81 ± 0.01	29.90 ± 0.97 i	177.55 ± 1.94 i
	2 A	62.25 ± 0.57 cd	5.81 ± 0.40	0.76 ± 0.03	29.65 ± 2.21 i	176.42 ± 2.52 ij
	2 B	53.51 ± 3.96 ef	6.47 ± 0.22	0.77 ± 0.02	37.91 ± 2.97 fgh	243.61 ± 3.80 g
30	Control A	17.45 ± 1.39 m	5.05 ± 0.86	0.89 ± 0.02	22.39 ± 0.81 klm	114.59 ± 1.88 m
	Control B	26.34 ± 1.14 kl	5.29 ± 0.40	0.88 ± 0.02	23.00 ± 1.47 jklm	117.35 ± 0.24 m
	1A	46.85 ± 0.46 g	5.91 ± 0.15	0.81 ± 0.01	36.73 ± 2.35 gh	206.66 ± 1.86 h
	1 B	32.11 ± 1.30 hij	5.78 ± 0.34	0.80 ± 0.03	38.22 ± 1.36 fgh	210.03 ± 3.49 h
	2 A	62.16 ± 1.93 cd	6.26 ± 0.32	0.70 ± 0.02	38.02 ± 2.69 fgh	237.42 ± 2.09 g
	2 B	36.85 ± 0.63 h	6.61 ± 0.24	0.71 ± 0.02	40.27 ± 0.76 efgh	256.17 ± 1.89 f
60	Control A	11.32 ± 1.65 no	6.85 ± 0.15	0.87 ± 0.07	20.90 ± 3.12 lmn	157.87 ± 2.66 k
	Control B	35.16 ± 1.04 h	6.90 ± 0.18	0.84 ± 0.03	24.70 ± 0.71 ijklm	165.73 ± 1.66 jk
	1A	66.44 ± 1.10 cd	7.14 ± 0.44	0.77 ± 0.01	43.36 ± 1.40 cdef	285.83 ± 4.77 e
	1 B	60.56 ± 0.34 de	7.49 ± 0.06	0.70 ± 0.01	42.46 ± 1.91 def	283.67 ± 2.31 e
	2 A	51.12 ± 1.76 fg	7.20 ± 0.15	0.70 ± 0.01	41.50 ± 0.85 defg	314.36 ± 3.39 d
	2 B	74.27 ± 1.81 b	6.86 ± 0.23	0.76 ± 0.03	49.66 ± 1.01 b	372.13 ± 2.97 b
90	Control A	29.43 ± 1.00 ijkl	6.36 ± 0.79	0.82 ± 0.02	23.13 ± 0.31 jklm	159.13 ± 5.34 k
	Control B	54.60 ± 0.99 ef	6.90 ± 0.11	0.83 ± 0.04	29.17 ± 1.11 i	200.12 ± 4.55 h
	1A	45.72 ± 2.10 g	7.17 ± 0.32	0.74 ± 0.04	45.23 ± 0.60 bcde	332.35 ± 3.61 c
	1 B	58.08 ± 1.59 de	7.00 ± 0.19	0.79 ± 0.05	48.01 ± 1.73 bc	342.25 ± 4.40 c
	2 A	51.04 ± 1.27 fg	7.57 ± 0.32	0.70 ± 0.01	46.21 ± 1.20 bcd	362.72 ± 4.90 b
	2 B	80.69 ± 0.02 a	7.41 ± 0.11	0.71 ± 0.01	57.97 ± 0.04 a	425.86 ± 1.38 a

Different letters in the same column indicate significant differences (*p* ≤ 0.01).

**Table 7 foods-09-00752-t007:** Analysis of variance of the chemical and color parameters studied on the loaves (*p*-values).

Factors of Variability	Degrees of Freedom	a_w_	Moisture	pH	HMF		Crumb			Crust	
						*L**	*a**	*b**	*L**	*a**	*b**
Type (A)	5	0.000	0.000	0.000	0.000	0.005	0.000	0.000	0.000	0.000	0.000
Storage time (B)	4	0.000	0.032	0.000	0.000	0.020	0.000	0.000	0.009	0.000	0.116
A × B	20	0.000	0.000	0.000	0.000	0.011	0.033	0.069	0.002	0.007	0.133

**Table 8 foods-09-00752-t008:** Evaluation of the chemical characteristics of the bread samples produced using different types and levels of sea salt during storage (data are means ± standard deviations).

Days of Storage	Type	a_w_	Moisture (%)	pH	HMF (mg/kg Dry Matter)
0	Control A	0.92 ± 0.00 abc	35.5 ± 0.07 n	5.65 ± 0.01 i	28.9 ± 1.73 fghi
	Control B	0.92 ± 0.00 a	37.1 ± 0.02 g	5.81 ± 0.00 efg	32.6 ± 2.18 de
	1A	0.92 ± 0.00 ab	36.1 ± 0.05 l	5.88 ± 0.01 cde	34.3 ± 1.40 cd
	1B	0.92 ± 0.00 ab	38.4 ± 0.04 b	5.94 ± 0.01 abc	38.2 ± 1.19 b
	2A	0.93 ± 0.00 a	35.8 ± 0.07 m	5.36 ± 0.00 l	39.2 ± 2.81 b
	2B	0.92 ± 0.00 a	36.7 ± 0.07 h	5.93 ± 0.02 abc	23.0 ± 1.07 mnopqr
15	Control A	0.91 ± 0.00 bcdef	36.6 ± 0.02 hi	5.80 ± 0.00 g	37.6 ± 0.58 bc
	Control B	0.92 ± 0.00 abcde	35.7 ± 0.08 mn	5.88 ± 0.01 cdef	29.2 ± 0.51 efgh
	1A	0.92 ± 0.00 abcd	39.5 ± 0.04 a	5.92 ± 0.01 abc	30.3 ± 0.98 efg
	1B	0.92 ± 0.00 abcd	37.9 ± 0.06 cd	5.93 ± 0.02 abc	30.8 ± 1.61 def
	2A	0.92 ± 0.00 abcd	37.12 ± 0.04 g	5.92 ± 0.01 abc	29.2 ± 0.64 efgh
	2B	0.92 ± 0.01 abcd	36.7 ± 0.08 h	5.96 ± 0.01 ab	27.9 ± 1.37 fghij
30	Control A	0.91 ± 0.00 defgh	35.9 ± 0.04 lm	5.73 ± 0.01 h	24.9 ± 0.27 jklmnop
	Control B	0.91 ± 0.01 cdefg	35.8 ± 0.02 m	5.78 ± 0.00 gh	21.0 ± 0.14 qr
	1A	0.92 ± 0.00 abcd	37.7 ± 0.05 de	5.99 ± 0.04 a	21.5 ± 0.03 pqr
	1B	0.92 ± 0.00 abcde	36.4 ± 0.07 i	5.88 ± 0.02 cdef	24.1 ± 1.14 lmnopqr
	2A	0.92 ± 0.00 abcd	37.3 ± 0.04 fg	5.84 ± 0.02 defg	37.8 ± 0.09 bc
	2B	0.92 ± 0.00 abcd	35.9 ± 0.03 lm	5.93 ± 0.02 abc	16.3 ± 0.03 s
60	Control A	0.90 ± 0.00 j	37.3 ± 0.07 fg	5.82 ± 0.03 efg	45.8 ± 0.07 a
	Control B	0.90 ± 0.00 ghij	35.0 ± 0.02 p	5.81 ± 0.01 fg	22.4 ± 0.20 nopqr
	1A	0.91 ± 0.00 efghij	37.5 ± 0.16 ef	5.83 ± 0.01 efg	27.7 ± 0.05 fghijk
	1B	0.91 ± 0.00 efghij	38.1 ± 0.06 c	5.78 ± 0.01 gh	26.0 ± 0.15 hijklm
	2A	0.91 ± 0.00 defghi	36.1 ± 0.02 l	5.83 ± 0.00 efg	25.0 ± 0.03 jklmno
	2B	0.91 ± 0.00 defgh	38.1 ± 0.08 c	5.91 ± 0.01 bcd	26.8 ± 0.08 ghijkl
90	Control A	0.88 ± 0.01 k	34.8 ± 0.06 p	5.73 ± 0.04 h	25.5 ± 0.13 ijklmn
	Control B	0.90 ± 0.00 ij	34.2 ± 0.02 q	5.81 ± 0.01 fg	24.2 ± 0.30 klmnopq
	1A	0.90 ± 0.00 hij	35.3 ± 0.07 o	5.78 ± 0.00 gh	20.6 ± 0.13 r
	1B	0.90 ± 0.00 fghij	34.2 ± 0.01 q	5.82 ± 0.08 efg	22.6 ± 0.03 mnopqr
	2A	0.90 ± 0.00 fghij	32.4 ± 0.04 r	5.80 ± 0.01 gh	21.5 ± 0.27 opqr
	2B	0.90 ± 0.00 fghij	32.4 ± 0.01 r	5.82 ± 0.00 efg	21.8 ± 0.00 opqr

Different letters in the same column indicate significant difference (*p* ≤ 0.01).

**Table 9 foods-09-00752-t009:** Influence of type of bread (6) on the attributes and mean scores of the significant sensory attributes (comparison of formulations). Data expressed as means.

Days of Storage	Attributes	F Values	Type					
			Control A	Control B	1A	1B	2A	2B
0	Salty	12.08 ***	4.2b	4.3b	1.6a	1.6a	1.6a	1.5a
15	Sweet	5.23 ***	2.9a	3.0a	4.3ab	5.2bc	5.8c	5.0bc
	Salty	7.49 ***	4.4b	4.1b	1.9a	1.9a	1.4a	1.7a
	Bread flavor	2.98 *	6.1b	6.2b	4.9ab	4.1a	4.6ab	3.6a
	Overall evaluation	5.01 ***	6.5b	6.3b	4.1a	3.9a	4.2a	3.4a
30	Sweet	5.30 ***	3.1ab	2.8a	4.3abc	4.6bc	6.4d	5.3cd
	Salty	7.40 ***	3.9b	4.2b	2.4a	2.2a	1.6a	1.7a
	Bread flavor	2.45 **	5.4b	5.5b	4.6ab	3.8ab	3.3a	3.5a
	Overall evaluation	3.48 **	5.8b	5.7b	4.6ab	3.8a	3.3a	3.6a
60	Sweet	3.25 *	2.9a	3.4a	4.1ab	4.4ab	5.6b	5.0b
	Salty	9.45 ***	5.2b	4.8b	2.8b	3.2b	1.9ab	1.3a
	Overall	3.17 *	5.4bc	5.6c	4.0ab	4.3abc	3.7a	3.2a
90	Sweet	6.45 ***	5.4bc	5.6c	3.8ab	4.1abc	3.5a	3.3a
	Salty	12.45 ***	5.0b	4.5b	2.3a	2.4a	1.8a	2.4a
	Overall evaluation	2.87 *	5.4bc	5.6c	3.8ab	4.1abc	3.5a	3.3a

Different letters in the same row indicate significant differences at *p* ≤ 0.05 *, *p* ≤ 0.01 **, *p* ≤ 0.001 ***.

**Table 10 foods-09-00752-t010:** Mean values of the significantly different sensory attributes (comparison during storage). Three bread loaves were collected at each sampling.

Attribute	Days of Storage	Control A	Control B	1A	1B	2A	2B
Elasticity	0 15 30 60 90		6.1 ab 7.5 b 6.5 ab 5.0 a 5.5 a				
Humidity	0 15 30 60 90	7.2 b 6.7 b 4.3 a 4.4 a 4.8 a	6.5 b 6.1 b 5.6 b 3.8 a 5.2 ab			7.2 b 5.7 ab 5.1 a 4.0 a 4.4 a	
Softness	0 15 30 60 90	6.5 bc 6.8 c 5.2 abc 4.4 a 4.9 ab	6.3 b 6.8 b 5.4 ab 4.2 a 5.4 ab				

Different letters in the same column indicate significant difference at *p* ≤ 0.05.

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
