# Peer review of "Development of Durum Wheat Breads Low in Sodium Using a Natural Low-Sodium Sea Salt"

_foods, 2020, doi:10.3390/foods9060752_

Round 1

Reviewer 1 Report

In the manuscript titled “Industrial development of … long storage” the authors Arena and co-workers performed experiments to assess the suitability of low sodium sea salt in preparing bread from durum wheat. This study included appropriate experimental designs and statistical tests on the data. An array of bread quality has been assessed including physio-chemical, texture, etc. The figures and table adequately depict the data. I don't have any major comments on this version of the manuscript. There are opportunities for improvement in the clarity of the text and presentation that could be attempted with structural changes, and english grammar and sentence compositions.

Minor comments:           

  1. The title appears too long and wordy. What would “functional” indicate here ?
  2. The introductory paragraph needs to be re-written to be more succinct and condensed to 3-4 paragraphs. For instance, line 42-68- pertaining to all food regulations could be condensed into one paragraph.
  3. Please provide expanded and abbreviations at the first occurrence in the manuscript. For instance: HMF

Reviewer 2 Report

“Industrial Development of Functional Durum Wheat Breads by Use of Sea Salt Low in Sodium: Evaluation of Quality Parameters During Long Storage” is a well-designed paper. The effects of different levels of traditional sea salt and reducing Na salt on the characteristics of re-milled semolina bread were investigated. Results are well summarized and discussed with the literature knowledge.

line 180: My advice is to provide some information about the remilled semolina particle size. In order to have a complete characterization of the raw material used for the study

line 182: It would be better to show the name and the year of the reference 

line 203: Please, check that the abbreviations have been described (HMF) and make sure in the entire manuscript.

line 218: My suggestion is to use cursive font for the CIELAB space color parameters

line 289: Please fix :-

line 471: The last column of the table results hard to read.

Reviewer 3 Report

  • The manuscript contains a lot of results - part of them is insignificant. In my opinion, this part of results should be transfered to Supplementary Material.
  • The article is not the place to look for the best statistical method to prove that two types of salt affect the bread quality. The Authors should decide which statistical method gives them the best results and only these results presented in the manuscript.
  • Discussion of the results should be improved, because the manuscript resembles a research report.
  • Section 2.2.1 - Provide more details about methods that you used to determine all mentioned parameters not only one reference.
  • Section 2.2.3.1 - ICP-OES analysis requires sample mineralization before the measurements. Did you mineralize your samples? Provide conditions of sample mineraliztion.
  • L.214 - What do you mean "rapeseed displacement"?
  • Section 2.4 - What did you determine usinh HPLC? What is the "HMF"?
  • L.274 - What microbiological attributes did you determine in the present studies?
  • Section 2.6 - Why did you use three different statistical softwares? Statistical methods, which you used, are the standard methods present in most, if not in all, statistical softwares.
  • Table 3 - Lack of statistical analysis.
  • Table3 and L.313-319 - Please decide how much salt you add to your breads. In Table 1, you provided other values.
  • L.311-312 - This should be in Materials and methods. Explain why did you calculate the Na concentration in thsi way.
  • L.317-319 - This senetnce should be rewritten.
  • Section 3.3 - In my opinion, Tables 5 and 6 ahould be presented before Table 4.
  • L. 369-371 - Provide references to this assumption.
  • L.435-438 - Explain what does 'the decraese in the HMF' mean for the bread quality and consumers? Provide discussion of the results with literature data.
  • L.528 - What does 'the convex hulls' mean?
  • Figures 1 and 2 are illegible.

Round 2

Reviewer 3 Report

The rewieved manuscript can be accepted for publication in Foods.